# MVI-Bench: A Comprehensive Benchmark for Evaluating Robustness to Misleading Visual Inputs in LVLMs

**Huiyi Chen** [1]  **Jiawei Peng** [* 2]  **Dehai Min** [* 1]  **Changchang Sun** [* 1]  **Kaijie Chen** [3]  **Yan Yan** [1]  **Xu Yang** [2]  **Lu Cheng** [1]

https://github.com/chenyil6/MVI-Bench

## Abstract

Evaluating the robustness of Large Vision-Language Models (LVLMs) is essential for their continued development and responsible deployment. However, existing robustness benchmarks largely focus on hallucination or misleading textual inputs, overlooking the critical challenge posed by misleading visual inputs in assessing visual understanding. To fill this gap, we introduce MVI-Bench, the first comprehensive benchmark specially designed for evaluating how Misleading Visual Inputs undermine the robustness of LVLMs. Grounded in fundamental visual primitives, the design of MVI-Bench centers on three hierarchical levels of misleading visual inputs: Visual Concept, Visual Attribute, and Visual Relationship. Using this taxonomy, we curate six representative categories and compile 1,248 expertly annotated VQA instances. To facilitate fine-grained robustness evaluation, we further introduce MVI-Sensitivity, a novel metric that characterizes LVLM robustness. Empirical results across 18 state-of-the-art LVLMs uncover pronounced vulnerabilities to misleading visual inputs, and our in-depth analyses on MVI-Bench provide actionable insights that can guide the development of more reliable and robust LVLMs.

## 1. Introduction

Recent advances in Large Vision-Language Models (LVLMs) (Yin et al., 2024) have driven remarkable progress

*Equal contribution  [1]Department of Computer Science, University of Illinois Chicago, Chicago, USA  [2]School of Computer Science & Engineering, Southeast University, Nanjing, China  [3]Guohao School, Tongji University, Shanghai, China. Correspondence to: Huiyi Chen <kitty22hy@gmail.com>, Lu Cheng <lucheng@uic.edu>.

*Proceedings of the 43rd International Conference on Machine Learning*, Seoul, South Korea. PMLR 306, 2026. Copyright 2026 by the author(s).

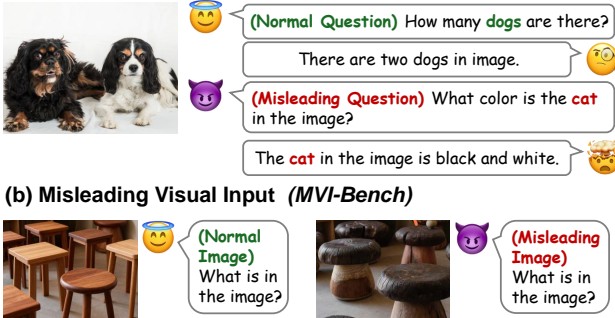

**(a) Misleading Textual Input** *(MAD-Bench)*

**(b) Misleading Visual Input** *(MVI-Bench)*

*Figure 1.* (a) **Misleading Textual Input**: misleading questions are created by injecting inaccurate or irrelevant information into otherwise normal queries. (b) **Misleading Visual Input**: misleading visual cues arise from real-world scenes, causing models to misinterpret the image content (*e.g.*, stools mistaken for mushrooms).

across a wide range of multimodal tasks, including image understanding (Chen et al., 2015; Wang et al., 2024a; Yang et al., 2023), visual question answering (Antol et al., 2015; Kuang et al., 2025; Li et al., 2024c; Peng et al., 2024), and complex visual reasoning (Wang et al., 2024c; Yue et al., 2024). With these rapid developments comes an urgent need for rigorous evaluation of LVLM robustness, motivating the creation of numerous robustness benchmarks (Fu et al., 2023; Agarwal et al., 2025; Jiang et al., 2025a).

While existing robustness benchmarks have provided valuable insights, they largely focus on specific challenges, such as hallucination (Bai et al., 2024; Wang et al., 2023) or adversarial robustness (Jiang et al., 2025a; Agarwal et al., 2025), and thus offer only a partial view of LVLM robustness. To broaden this scope, recent efforts (Dang et al., 2024; Deng et al., 2025; Qian et al., 2024) have begun exploring LVLM behavior under misleading scenarios, typically by introducing misleading textual inputs. For instance, MAD-Bench (Qian et al., 2024) constructs misleading questions by injecting inaccurate information into textual prompts (Fig. 1(a)). However, misleading information is not confined to the language modality. In real-world settings, misleading

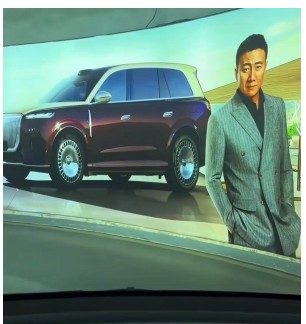 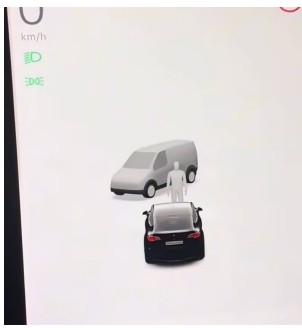

| **(a) Advertisement billboard** | **(b) Vehicle's perception view** |

*Figure 2.* A real-world example of misleading visual inputs that deceive machines: the vehicle's perception system misidentifies the 2D content on an advertisement billboard (a) as actual road objects (b), triggering unnecessary emergency braking.

visual inputs naturally arise and can deceive both machines (see Fig. 2) and humans.

This phenomenon has long been examined in philosophy under the notion of the Problem of Perception (Smith, 2005), which emphasizes that human visual cognition is prone to systematic errors induced by perceptual biases. As illustrated in Fig. 1(b), humans may initially misinterpret visually ambiguous scenes, such as perceiving stools as mushrooms, but can often correct these misconceptions through contextual reasoning and prior knowledge.

This observation reveals a fundamental challenge for LVLMs: unlike humans, who can correct initial misperceptions through contextual reasoning and prior knowledge, models often rely on superficial visual patterns and struggle to disambiguate misleading cues. Yet, this critical dimension remains underexplored. Moreover, prior works on misleading scenarios typically focus on textual inputs with deliberately injected misinformation, which rarely occurs in real-world interactions and differs fundamentally from naturally arising visual misleading cues.

This gap motivates us to introduce the **M**isleading **V**isual **I**nput **Bench**mark (**MVI-Bench**), the first comprehensive benchmark specifically designed to evaluate the robustness of LVLMs against misleading visual inputs. Grounded in fundamental visual primitives (Han et al., 2025), the design of MVI-Bench is organized around three hierarchical levels of misleading cues: *Visual Concept, Visual Attribute,* and *Visual Relationship*. Each level further comprises several representative misleading categories that closely mirror the types of perceptual errors humans encounter in real-world visual experiences, as detailed in Section 3.1. MVI-Bench includes a total of 624 pairs of VQA instances (*i.e.*, 1,248 instances in total), where each pair consists of a normal image and its corresponding misleading counterpart. The two images within a pair share nearly identical semantic

content, differing only by the introduction of subtle misleading visual cues (see Fig. 3 for details). To complement this paired design, we further propose **MVI-Sensitivity**, a novel evaluation metric that quantifies the relative performance degradation from normal to misleading visual inputs. This metric provides a fine-grained measure of LVLM robustness, enabling a deeper understanding of model sensitivity to misleading visual inputs.

In summary, our main contributions are as follows:

- We establish the first comprehensive taxonomy of misleading visual inputs. Grounded in fundamental visual primitives, this taxonomy systematically characterizes Visual Concept, Visual Attribute, and Visual Relationship as the core dimensions for evaluating LVLM behavior under misleading visual conditions.
- We introduce MVI-Bench, a carefully curated benchmark comprising six representative categories and 624 expertly annotated VQA instance pairs. Each pair is designed to isolate the effects of misleading visual cues while preserving the underlying semantic content.
- We propose MVI-Sensitivity, a novel evaluation metric that enables fine-grained robustness assessment by quantitatively measuring the performance degradation of LVLMs when exposed to misleading visual inputs.
- Through extensive experiments and in-depth analyses on 18 open- and closed-source LVLMs, we uncover three key observations: limited LVLMs robustness, visual perception and reasoning are complementary, and the presence of spurious correlation. These findings offer actionable insights for developing more reliable and robust LVLMs.

## 2. Related Work

**Large Vision-Language Models (LVLMs).** LVLMs unify visual perception and visual reasoning through an architecture comprising a vision encoder, a large language model (LLM), and a connector module that aligns visual and textual feature spaces, enabling seamless multimodal understanding (Awadalla et al., 2023; Dai et al., 2023; Bai et al., 2023; Liu et al., 2023; Zhu et al., 2023; Dong et al., 2025; Abdin et al., 2024; Chen et al., 2024; Team et al., 2025). Leveraging this architecture, LVLMs have rapidly evolved from early multimodal frameworks such as Flamingo (Awadalla et al., 2023) and InstructBLIP (Dai et al., 2023) to a new generation of high-performing models, such as LLaVA (Liu et al., 2023) and Qwen-VL (Bai et al., 2023). When deployed in real-world settings, LVLMs exhibit remarkable advancements across multiple multimodal tasks, including image understanding (Chen et al., 2015; Wang et al., 2024a; Chen et al., 2025), visual question answering (Antol et al., 2015; Kuang et al., 2025; Jiang et al., 2025b), and complex visual reasoning (Wang et al., 2024c;

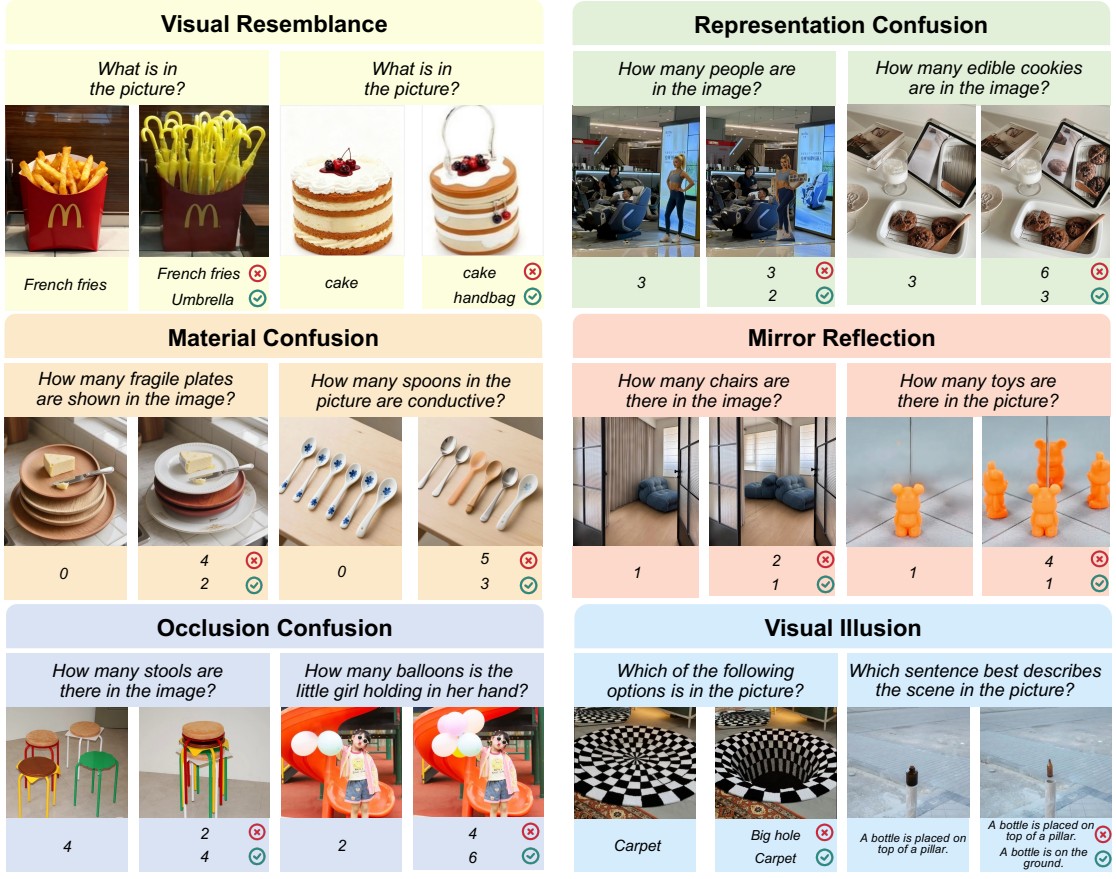

*Figure 3.* Examples from six misleading categories defined in MVI-Bench. Each pair contains a *normal image* (left) and *misleading image* (right) with the same MCQ and corresponding ground-truth answer. Answer choices are omitted for brevity (see Fig. 5 for full format).

Yue et al., 2024; Fu et al., 2026). This progress is reflected across both open-source and proprietary LVLM families. For example, open-source series such as InternVL (Chen et al., 2024) and SAIL-VL (Yin et al., 2025) have made substantial progress, while proprietary models like GPT-4 (Achiam et al., 2023) and Gemini (Team et al., 2023) further push the frontier through large-scale multimodal training and enhanced reasoning capabilities.

**Robustness Benchmarks for LVLMs.** Existing studies on LVLM robustness have primarily focused on two perspectives. On the one hand, prior works examine hallucination issues, which evaluates the discrepancy between generated responses and the visual content (Bai et al., 2024; Wang et al., 2023; Jiang et al., 2024). On the other hand, a separate line of research investigates adversarial vulnerabilities, analyzing how models behave under carefully crafted perturbated inputs (Jiang et al., 2025a; Agarwal et al., 2025; Zhao et al., 2023). However, these evaluations illuminate only a partial view of LVLM robustness. To broaden this scope, recent researches have begun to investigate LVLM behavior under misleading scenarios, most commonly through misleading textual inputs (Liu et al., 2025b; Dang et al.,

2024; Deng et al., 2025; Qian et al., 2024). However, robustness against misleading visual inputs, a challenge that naturally arises in real-world environments, remains substantially underexplored. Although a few recent efforts, such as IllusionVQA (Shahgir et al., 2024) and O-Bench (Liu et al., 2025c), have begun to investigate this direction, they focus on a single type of misleading phenomenon and rely on coarse metrics such as accuracy, falling short of fine-grained robustness analysis. MVI-Bench addresses these limitations through comprehensive category coverage and a paired evaluation design, enabling unique insights unattainable from prior benchmarks.

## 3. MVI-Bench

To comprehensively evaluate LVLM robustness under misleading visual inputs, we propose MVI-Bench, a paired VQA benchmark. Specifically, each instance pair shares the same multiple-choice question (MCQ) with four options (A-D) but differs in visual input: a *normal image* and its *misleading counterpart* (Fig. 3). The two images are visually similar in overall composition, but the misleading one

*Table 1.* **Comparison between MVI-Bench and prior visually misleading benchmarks.** MVI-Bench uniquely covers six well-defined misleading categories and includes diverse image sources with a paired design, enabling comprehensive and controlled robustness evaluation. (Abbreviations: Res.-Visual resemblance; Rep.-Representation confusion; Mat.-Material confusion; Mir.-Mirror reflection; Occ.-Occlusion confusion; Ill.-Visual illusion.)

| Dataset | Venue | #Image | Res. | Rep. | Mat. | Mir. | Occ. | Ill. | Image Source | | | Paired |
| --- | --- | --- | --- | --- | --- | --- | --- | --- | --- | --- | --- | --- |
| | | | | | | | | | Natural | Synthetic | Edited | |
| IllusionVQA (Shahgir et al., 2024) | COLM 2024 | 374 | ✗ | ✗ | ✗ | ✗ | ✗ | ✓ | ✓ | ✗ | ✗ | ✗ |
| IllusionBench+ (Zhang et al., 2025b) | arXiv 2025.6 | 1,051 | ✗ | ✗ | ✗ | ✗ | ✗ | ✓ | ✓ | ✗ | ✗ | ✗ |
| O-Bench (Liu et al., 2025c) | arXiv 2025.8 | 1,365 | ✗ | ✗ | ✗ | ✗ | ✓ | ✗ | ✗ | ✓ | ✗ | ✗ |
| **MVI-Bench (ours)** | – | **1,248** | ✓ | ✓ | ✓ | ✓ | ✓ | ✓ | ✓ | ✓ | ✓ | ✓ |

introduces subtle cues designed to confuse the model. This paired design enables **controlled analysis** of how visual misleading affects LVLMs' behavior: for each image, only one option is correct, and at least one distractor is intentionally crafted to appear plausible given the misleading visual cues. Overall, MVI-Bench comprises 624 such pairs (1,248 instances total) across six categories, enabling systematic evaluation under visually misleading conditions. Tab. 1 presents a detailed comparison of MVI-Bench with existing related benchmarks.

### 3.1. Taxonomy of Visually Misleading Types

We ground our taxonomy in the principles of fundamental visual primitives (Han et al., 2025), namely, *Visual Concept, Visual Attribute,* and *Visual Relationship*[1]. These primitives form the core of visual cognition and offer a structured lens through which misleading phenomena can be characterized. Accordingly, we define three corresponding levels of misleading visual inputs and their associated categories.

**1. Visual Concept Misleading.** Misleading effects at this level arise from coarse-grained visual similarity between semantically different entities (*e.g.*, objects, people, scenes). Addressing such misleading inputs requires finer visual discrimination and modest reasoning that goes beyond superficial pattern matching. This level includes two categories:

- *Visual Resemblance.* Cases where models may confuse an object with a visually similar but semantically different object or an object from a different category.
- *Representation Confusion.* Cases where models can fail to distinguish between an actual object and its two-dimensional representation (*e.g.*, photograph or painting).

**2. Visual Attribute Misleading.** This level captures misleading cases from fine-grained ambiguity in visual attributes (*e.g.*, texture, gloss, material). Correctly resolving these cases requires refined perceptual discrimination to differentiate materials that may look similar at a coarse level but differ fundamentally. This level includes one category:

- *Material confusion.* Cases where models sometimes confuse items based on their texture, material, or other physical attributes.

**3. Visual Relationship Misleading.** This level encompasses misleading cases that arise from incorrect interpretation of visual relationships, which describe spatial arrangements, part–whole structures, or contextual dependencies among entities. Unlike concept- or attribute-level misleading, relationship-level misleading requires the model to reason about higher-order spatial arrangements and interactions between objects. This level includes three categories:

- *Mirror Reflection.* Cases where models can be misled by virtual images from mirror reflections, confusing them with actual, real-world objects.
- *Occlusion Confusion.* Cases where models struggle to recognize an object when part of it is visually occluded, sometimes leading to incorrect identification.
- *Visual Illusion.* Cases where models are susceptible to visual illusions arising from complex spatial arrangements, lighting conditions, or deceptive perspectives.

Above all, these three levels and six categories form a comprehensive taxonomy for evaluating LVLMs' susceptibility to misleading visual inputs. This taxonomy spans the full spectrum of visual understanding from low-level recognition and mid-level attribute discrimination to high-level spatial and contextual reasoning. Representative examples from each category are illustrated in Fig. 3.

### 3.2. Data Curation

**Data Collection.** Following the taxonomy of visually misleading types described above, three trained human experts with at least a bachelor's degree manually curated an initial set of image pairs from three complementary sources to provide broad topical coverage:

(1) **Natural images**, collected from multiple international social media platforms to ensure broad and diverse real-world content coverage;

(2) **Synthetic images**, generated using the generative model Seedream (Seedream et al., 2025) to supplement misleading

---

[1]We exclude "visual combination", since our aim is to isolate and analyze the effect of each individual misleading level.

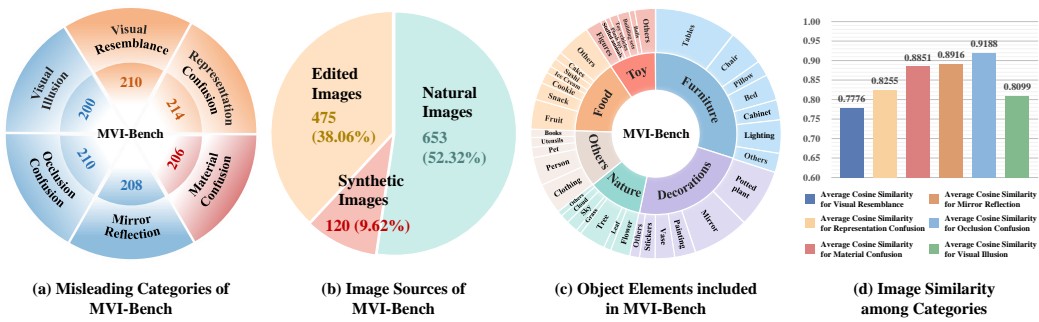

*Figure 4.* Overview of MVI-Bench statistics.

cases that are rare and low-quality in natural data;

(3) **Edited images**, created by human experts using advanced image editing tools to modify natural or synthetic images, typically by removing misleading visual cues to serve as normal counterparts.

Normal images in our benchmark primarily originate from natural (23.56%) and edited images (76.44%), while misleading images are mainly sourced from natural (81.09%) and synthetic ones (18.91%). For each image, human experts craft a MCQ based on its visual content, with exactly one correct answer and at least one distractor that plausibly leverages the misleading visual cues. Additionally, each annotated VQA pair including its label ("misleading" or "normal") and the answer is independently reviewed by additional human experts, and any controversial cases are discarded to ensure quality.

**Data Filtering and Refinement.** To further ensure the discriminative value of MVI-Bench, we evaluate the initially collected data using two strong LVLMs: InternVL-3-2B (Chen et al., 2024) and Qwen2.5-VL-3B (Bai et al., 2025). Based on the evaluation results, we then identify instances where both models correctly answer the questions for both normal and misleading images. Since these instances pose little challenge and provide limited ability to differentiate model robustness, we treat them as low-difficulty samples. To maintain dataset diversity while preventing such easy cases from dominating the benchmark, we randomly retain only a portion of them. After this refinement process, the dataset is reduced from 1,578 to 1,248 high-quality, discriminative VQA instances.

**Data Statistics.** Our final benchmark comprises 1,248 VQA instances. Fig. 4 illustrates the overall statistics of MVI-Bench from four complementary perspectives. Fig. 4 (a) shows the data distribution across our taxonomy, with roughly balanced instances per category for unbiased evaluation. Fig. 4 (b) depicts the composition of our three image sources: natural images (52.32%), synthetic images (9.62%), and edited images (38.06%). Fig. 4 (c) shows that the benchmark exhibits broad semantic coverage, en-

compassing a wide range of categories such as food, furniture, nature, and decorations. Moreover, each normal and misleading image pair is meticulously designed to isolate the effect of visual misleading cues while preserving other semantic content, ensuring minimal differences between paired images. Fig. 4 (d) shows empirical evidence for this design by showing high cosine similarity (using features obtained using CLIP-Large (Radford et al., 2021)) between paired images across all six categories.

### 3.3. Evaluation Metrics

We employ two evaluation metrics. First, we report **Accuracy** on normal and misleading examples to directly measure model performance under subtly different visual conditions. Second, we introduce **MVI-Sensitivity**, a new metric designed to quantify the extent to which an LVLM's performance degrades when exposed to misleading visual inputs, compared to corresponding normal images:

$$\text{MVI-Sensitivity} = \frac{|Acc_\text{n} - Acc_\text{m}|}{Acc_\text{n}}, \quad (1)$$

where $Acc_\text{n}$ and $Acc_\text{m}$ denote the accuracy on normal and misleading samples, respectively. A lower MVI-sensitivity indicates that the model is less affected by misleading cues and exhibits stronger robustness.

## 4. Experiments

We employ MVI-Bench to comprehensively study LVLMs' behavior under misleading visual inputs. Our experimental design centers on the following research questions:

- **RQ1**: How do LVLMs perform on MVI-Bench across different categories of misleading visual inputs?
- **RQ2**: How do visual perception and reasoning abilities contribute respectively to LVLMs robustness in visually misleading scenarios?
- **RQ3**: What insights can be derived from counterintuitive cases where LVLMs behave unexpectedly under misleading visual inputs?

## 4.1. Models and Experimental Settings

We evaluate 18 state-of-the-art LVLMs on MVI-Bench to systematically assess their robustness against misleading visual inputs. These models span a diverse range of architectures and scales. Specifically, it includes **six closed-source models**: GPT-4o (Hurst et al., 2024), GPT-4.1 (Achiam et al., 2023), Claude-3.7-Sonnet (Anthropic, 2025), Gemini-2.5-Flash (Team et al., 2023), Gemini-2.5-Pro (Team et al., 2023) and GPT-5-Chat (OpenAI, 2025); and **twelve open-source models**, including Qwen2.5-VL series (Bai et al., 2025), the InternVL-3 series (Zhu et al., 2025a), the SAIL-VL2 series (Yin et al., 2025), LLaVA-OneVision (Li et al., 2024b) and Molmo (Deitke et al., 2025). For closed-source models, the temperature is fixed at 0.0, and for open-source models, the decoding is performed using a greedy strategy. Meanwhile, modern LLMs are known to be vulnerable to option-position variations in MCQs due to their inherent 'selection bias' (Zheng et al., 2023), namely a tendency to prefer specific option IDs as answers. To mitigate this bias, we randomly shuffle the answer options each time the LVLMs are queried.

## 4.2. Main Results (RQ1)

We report the evaluation results of 18 LVLMs in Tab. 2 across 6 misleading categories in MVI-Bench. The main findings are summarized as follows.

**Existing LVLMs remain highly susceptible to misleading visual inputs, especially open-source ones.** As shown in Tab. 2, all evaluated LVLMs exhibit substantial performance degradation when exposed to misleading visual cues. On the closed-source side, even the latest state-of-the-art models are not immune. For instance, GPT-5-Chat's accuracy drops from 90.00% to 61.00% when *visual illusion* is introduced, and Gemini-2.5-Pro's accuracy decreases from 86.54% to 54.81% under *mirror reflection*. All closed-source LVLMs yield overall MVI-Sensitivity above 20%, indicating substantial vulnerability. Notably, Claude-3.7-Sonnet, despite its strength on reasoning-intensive tasks such as coding, performs the worst among closed-source models, achieving only 42.13% accuracy on misleading images (Acc$_m$) and an MVI-Sensitivity of 42.10%.

In contrast, open-source models demonstrate greater susceptibility. The strongest, Qwen2-VL-72B, reaches 32.38% MVI-Sensitivity, yet lags noticeably behind top closed-source counterparts. Molmo-7B, a fully-open model releasing both its training data and recipes, performs worst at 48.69% MVI-Sensitivity, meaning nearly half of its responses are affected by misleading cues. The performance gap between open- and closed-source models may stem from several factors (Li et al., 2024a; Laurençon et al., 2024; Chen et al., 2024): closed-source models typically benefit from larger-scale, higher-quality proprietary training data (Sun et al., 2024; Ye et al., 2025; Min et al., 2025) and more extensive computational budgets enabling sophisticated post-training refinement techniques (Yu et al., 2025; Li et al., 2024a).

**LVLMs demonstrate stronger mastery of coarse-grained visual concepts than fine-grained visual attributes.** Within the visual concept level, both *visual resemblance* and *representation confusion* involve coarse-grained semantic discrimination. LVLMs consistently show high and stable robustness on these two categories: most LVLMs models exceed or approach 50% Acc$_m$ in *visual resemblance*, and all models except Molmo maintain over 40% Acc$_m$ in *representation confusion*. These results indicates that current LVLMs handle high-level semantic discrimination more reliably than other types of misleading visual inputs. For example, Qwen2-VL-72B achieves highest 76.19% Acc$_m$ and 15.79% MVI-sensitivity in *visual resemblance*, and GPT-4.1 reaches an 74.77% Acc$_m$ with the lowest MVI-Sensitivity of 8.04% in *representation confusion*. However, performance drops considerably in *material confusion*, where the best model, Gemini-2.5-Pro, attains only 66.00% Acc$_m$ and GPT-4o falls even further to 43.00%.

**Spatial reasoning remains a critical weakness of current LVLMs.** *Mirror reflection* is common in everyday life, yet it remains highly challenging for LVLMs. Even Qwen2-VL-72B, achieves only 50.00% Acc$_m$, and GPT-4o reaches merely 52.88%. This weakness is more pronounced among smaller models: InternVL3-2B, Qwen2.5-VL-3B, and LLaVA-OneVision-7B all exhibit MVI-Sensitivity exceeding 70%. This indicates that while LVLMs perform well on images without mirror reflections (Acc$_n$ often exceeds 80%), their performance drops sharply once virtual reflections appear, suggesting fundamental difficulties in distinguishing real objects from mirror images.

Similarly, *occlusion confusion* reveals another spatial reasoning limitation. Only Gemini-2.5-Flash, Gemini-2.5-Pro and GPT-5-Chat surpass 50% Acc$_m$, while all other models fall below, highlighting a substantial deficiency in occlusion perception—a capability that underpins human spatial understanding. Notably, performance differences across models on *visual illusion* are less pronounced than in other categories. For example, InternVL3's Acc$_m$ varies only between 48% and 52% as model size increases from 2B to 78B. This may stem from the scarcity of visual illusion examples in multimodal pre-training datasets, suggesting targeted data augmentation as a promising direction.

## 4.3. Assessing the Impact of Visual Perception and Reasoning (RQ2)

To understand what drives robustness against misleading visual inputs, we separately investigate the impact of visual perception and visual reasoning capabilities through

*Table 2.* **Performance comparison on MVI-Bench.** Each category reports model and human performance on **normal** ($\text{Acc}_n$) and **misleading** ($\text{Acc}_m$) images, as well as **MVI-Sensitivity** (Sens). For each category and the overall results, the best score is in **bold** and the second best is underlined. $\text{Acc}_m$ evaluates performance under misleading visual cues (↑: higher is better), while Sens measures the relative performance drop from normal to misleading inputs (↓: lower is better).

| Misleading Level | Concept | | | | | | Attribute | | | Relationship | | | | | | | | | Overall | |
| --- | --- | --- | --- | --- | --- | --- | --- | --- | --- | --- | --- | --- | --- | --- | --- | --- | --- | --- | --- | --- |
| Misleading Type | Resemblance | | | Representation | | | Material | | | Mirror | | | Occlusion | | | Illusion | | | | |
| Model | $\text{Acc}_n \to \text{Acc}_m$ | | Sens | $\text{Acc}_n \to \text{Acc}_m$ | | Sens | $\text{Acc}_n \to \text{Acc}_m$ | | Sens | $\text{Acc}_n \to \text{Acc}_m$ | | Sens | $\text{Acc}_n \to \text{Acc}_m$ | | Sens | $\text{Acc}_n \to \text{Acc}_m$ | | Sens | $\text{Acc}_m$ | Sens |
| | | | | | | | | | | | | | | | | | | | ↑ | ↓ |
| Open-source Models | | | | | | | | | | | | | | | | | | | | |
| LLaVA-OV (7B) | 89.52 → 66.67 | | 25.53 | 82.24 → 50.47 | | 38.63 | 77.67 → 45.63 | | 41.25 | 88.46 → 23.08 | | 73.91 | 70.48 → 29.52 | | 58.12 | 87.00 → 47.00 | | 45.98 | 43.75 | 46.99 |
| Molmo (7B) | 82.52 → 49.52 | | 40.23 | 70.09 → 23.36 | | 66.67 | 65.05 → 34.95 | | 46.27 | 80.77 → 21.15 | | 73.81 | 62.86 → 48.57 | | 22.73 | 79.00 → 49.00 | | 37.97 | 37.66 | 48.69 |
| SAIL-VL2 (2B) | 91.43 → 53.33 | | 41.67 | 79.44 → 65.42 | | 17.65 | 73.79 → 46.60 | | 36.85 | 93.27 → 36.54 | | 60.85 | 68.57 → 39.05 | | 43.06 | 85.00 → 48.00 | | 43.53 | 48.23 | 41.10 |
| SAIL-VL2 (8B) | 90.48 → 57.14 | | 36.85 | 81.31 → 63.55 | | 21.84 | 72.82 → 45.63 | | 37.34 | 92.31 → 39.42 | | 57.29 | 72.38 → 38.10 | | 47.36 | 85.00 → 51.00 | | 40.00 | 49.19 | 40.28 |
| InternVL3 (2B) | 83.81 → 40.00 | | 52.27 | 70.09 → 45.79 | | 34.67 | 65.05 → 42.72 | | 34.33 | 86.54 → 25.96 | | 70.00 | 64.76 → 32.38 | | 50.00 | 84.00 → 51.00 | | 39.29 | 39.58 | 47.67 |
| InternVL3 (8B) | 88.57 → 60.95 | | 31.18 | 75.70 → 58.88 | | 22.22 | 71.84 → 45.75 | | 36.32 | 93.27 → 30.77 | | 67.01 | 67.62 → 41.90 | | 38.04 | 86.00 → 49.00 | | 43.02 | 48.23 | 40.04 |
| InternVL3 (78B) | 89.52 → 66.67 | | 25.53 | 83.18 → 64.49 | | 22.47 | 78.64 → 62.14 | | 20.98 | 89.42 → 47.12 | | 47.30 | 78.12 → 48.57 | | 37.83 | 86.00 → 52.00 | | 39.53 | 56.89 | 32.38 |
| Qwen2-VL (72B) | 90.48 → **76.19** | | 15.79 | 81.31 → 69.16 | | 14.94 | 84.47 → 51.46 | | 39.10 | 91.35 → 50.00 | | 45.26 | 75.24 → 40.95 | | 45.57 | 87.00 → 61.00 | | 29.89 | 58.17 | 31.52 |
| Qwen2.5-VL (3B) | 87.62 → 62.86 | | 28.26 | 71.96 → 49.53 | | 31.17 | 68.93 → 38.83 | | 43.67 | 84.62 → 23.08 | | 72.73 | 60.95 → 31.43 | | 48.43 | 79.00 → 43.00 | | 45.57 | 41.51 | 45.01 |
| Qwen2.5-VL (7B) | 87.62 → 53.33 | | 39.14 | 77.57 → 60.75 | | 21.68 | 77.67 → 46.60 | | 40.00 | 92.31 → 43.72 | | 52.64 | 71.43 → 29.52 | | 58.67 | 85.00 → 42.00 | | 50.59 | 45.99 | 43.84 |
| Qwen2.5-VL (32B) | 91.43 → 60.00 | | 34.38 | 79.44 → 59.81 | | 24.72 | 73.79 → 43.69 | | 40.78 | 89.42 → 37.50 | | 58.08 | 74.29 → 30.48 | | 58.97 | 87.00 → 54.00 | | 37.93 | 47.59 | 42.35 |
| Qwen2.5-VL (72B) | 92.38 → 63.81 | | 30.92 | 78.50 → 67.29 | | 14.28 | 79.61 → 51.46 | | 35.36 | 89.42 → 49.04 | | 45.16 | 80.95 → 48.57 | | 40.00 | 87.00 → **63.00** | | **27.57** | 57.21 | 32.38 |
| Closed-source Models | | | | | | | | | | | | | | | | | | | ↑ | ↓ |
| Claude-3.7-Sonnet | 90.48 → 54.29 | | 40.00 | 72.90 → 43.93 | | 39.74 | 59.22 → 38.83 | | 34.43 | 73.08 → 29.81 | | 59.21 | 57.14 → 39.05 | | 31.66 | 86.00 → 48.00 | | 44.19 | 42.13 | 42.10 |
| GPT-4o | 90.48 → 74.29 | | 17.89 | 71.96 → 54.21 | | 24.67 | 58.00 → 43.00 | | 25.86 | 77.88 → 52.88 | | 32.10 | 59.05 → 42.86 | | 27.42 | 85.00 → 54.00 | | 36.47 | 53.37 | 27.28 |
| GPT-4.1 | 86.67 → 73.33 | | **15.39** | 81.31 → **74.77** | | **8.04** | 68.00 → 55.00 | | 19.12 | 86.54 → 67.31 | | 22.22 | 67.62 → 49.52 | | 26.77 | 88.00 → 58.00 | | 34.09 | 62.82 | **20.80** |
| Gemini-2.5-Flash | 86.67 → 61.90 | | 28.58 | 72.90 → 49.50 | | 32.13 | 74.76 → 55.34 | | 25.98 | 72.12 → 36.54 | | 49.33 | 60.95 → 55.23 | | **9.38** | 80.00 → 54.00 | | 32.50 | 52.08 | 30.11 |
| Gemini-2.5-Pro | 90.48 → 67.62 | | 25.26 | 79.44 → 64.49 | | 18.82 | 81.00 → **66.00** | | **18.52** | 86.54 → 54.81 | | 36.67 | 78.10 → **65.71** | | 15.86 | 91.00 → 58.00 | | 36.26 | 62.50 | 22.00 |
| GPT-5 Chat | 83.81 → 70.48 | | 15.91 | 84.11 → 70.09 | | 16.67 | 84.47 → 59.22 | | 29.89 | 85.58 → **70.19** | | 17.98 | 69.52 → 51.43 | | 26.02 | 90.00 → 61.00 | | 32.22 | **63.78** | 23.02 |
| Human Performance | | | | | | | | | | | | | | | | | | | ↑ | ↓ |
| Human | 100.0 → 98.10 | | 1.99 | 100.0 → 98.43 | | 1.57 | 99.03 → 97.09 | | 1.96 | 100.0 → 98.08 | | 1.92 | 100.0 → 100.0 | | 0.00 | 100.0 → 98.00 | | 2.00 | 98.24 | 1.63 |

*Table 3.* Performance of Qwen2.5-VL-7B with caption-assisted inference. Baseline (caption model set to None) uses no caption.

| Caption Model | Res. | Rep. | Mat. | Mir. | Occ. | Ill. | Overall $\text{Acc}_m$ |
| --- | --- | --- | --- | --- | --- | --- | --- |
| None | 53.33 | 60.75 | 46.60 | 43.72 | 29.52 | 42.00 | 45.99 |
| Qwen2.5-VL | 54.29 | 61.68 | 40.78 | 40.38 | 26.67 | 52.00 | 45.99 (+0.0) |
| GPT-4.1 | 58.10 | 71.96 | 49.51 | 53.85 | 33.33 | 56.00 | 53.85 (+7.86) |

*Table 4.* Comparison of model performance without and with the thinking process across misleading visual categories.

| Model | Res. | Rep. | Mat. | Mir. | Occ. | Ill. | Overall $\text{Acc}_m$ |
| --- | --- | --- | --- | --- | --- | --- | --- |
| w/o thinking process | | | | | | | |
| SAIL-VL-2-8B | 57.14 | 63.55 | 45.63 | 39.42 | 38.10 | 51.00 | 49.19 |
| Qwen2-VL-72B | 76.19 | 69.16 | 51.46 | 50.00 | 40.95 | 61.00 | 58.17 |
| Gemini-2.5-Flash | 61.90 | 49.50 | 55.34 | 36.54 | 55.23 | 54.00 | 52.08 |
| GPT-5-Chat | 70.48 | 70.09 | 59.22 | 70.19 | 51.43 | 61.00 | 63.78 |
| w/ thinking process | | | | | | | |
| SAIL-VL2-8B | 49.52 | 55.14 | 54.37 | 47.12 | 34.29 | 53.00 | 48.08 (-1.11) |
| QVQ-72B-Preview | 61.90 | 66.36 | 54.81 | 21.90 | 45.00 | | 51.76 (-6.41) |
| Gemini-2.5-Flash | 64.76 | 72.90 | 65.05 | 63.46 | 59.05 | 57.00 | 63.14 (+11.06) |
| GPT-5-thinking | 67.62 | 73.81 | 66.99 | 57.69 | 55.24 | 65.00 | 64.42 (+0.64) |

controlled experiments, yielding following two key findings.

**Finding 1: Enhanced visual perception substantially improves robustness against misleading visual inputs.** The vision encoder serves as the "eye" of an LVLM, determining its perceptual upper bound (Tong et al., 2024; Neo et al., 2024; Zhang et al., 2025a). Enhancing the model's ability to perceive and encode visual details may improve its robustness. To test this hypothesis, we employ a *caption-assisted inference* approach (Wan et al., 2025; Liang et al., 2025), where image captions are used as a proxy for enhanced perception. Since captions describe *what is present in the image* rather than involving multi-step reasoning, they primarily capture perceptual information (Vinyals et al., 2015; Tong et al., 2024). Specifically, we use a strong auxiliary LVLM (the *caption model*) to generate a detailed image caption, which is then concatenated with the image and question as input to the *inference model* (see Appendix D.1 for the prompt format).

As shown in Tab. 3, when Qwen2.5-VL-7B answers directly without any additional captions, its $\text{Acc}_m$ is only 45.99%. When the model itself serves as the caption model, performance remains nearly unchanged. However, replacing the caption model with a stronger GPT-4.1 yields substantial gains, increasing overall $\text{Acc}_m$ from 45.99% to 53.85% with improvements across all categories. Notably, this performance surpasses Qwen2.5-VL-32B (47.59%, as shown in Tab. 2) and approaches the results of Qwen2.5-VL-72B (57.21%). These results suggest that the visual perceptual capacity may be a key factor of LVLM robustness under visually misleading conditions.

**Finding 2: Enhanced visual reasoning improves robust-**

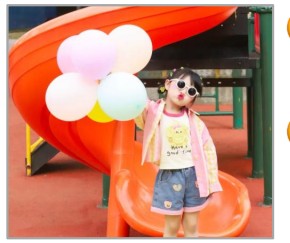

*(non-think)* The little girl **is holding six balloons in her hand**. The balloons are a mix of pastel colors, including pink, white, yellow, and light blue. They are clustered together in her hand, and the image clearly shows her holding them. So the final answer should be **B**. ✅

*(think)* <think>Got it, let's see. The question is asking how many balloons the little girl is holding. The options are A. Five, B. Six, C. Four, D. Seven.

First, I need to look at the image description provided. The description says, "holding a bunch of balloons" and then lists the **colors**: "white, pink, yellow, light blue, and green". Let me count those**. White is one, pink is two, yellow three, light blue four, green five. So that's five balloons.**

**Wait, the original COT mentioned the description lists five colors,** so that would be five. The answer was A. But let me make sure I didn't miss any. **The description says "a bunch of balloons" with colors: white, pink, yellow, light blue, green**. **That's five distinct colors, so five balloons**. So the correct answer should be **A**.</think> ❌

**How many balloons is the little girl holding in her hand?**
A. Five   **B. Six**
C. Four   D. Seven

*Figure 5.* **Comparison between the "non-think" and "think" modes of SAIL-VL.** In the non-think mode, the model answers directly based on visual evidence, while in the think mode, the model is guided by **historical thoughts** and tend to **overemphasize fine details**.

**ness against misleading visual inputs, yet the improvement remains inconsistent and unstable.** There are currently two mainstream approaches to strengthen visual reasoning in LVLMs: (1) scaling the language model (Wei et al., 2022a; Berti et al., 2025), and (2) enabling explicit reasoning via long-form chain-of-thought (CoT) (Wei et al., 2022b; Bi et al., 2025; Liu et al., 2026) through reinforcement learning (RL) (Guo et al., 2025; Peng et al., 2025; Wu et al., 2025). We examine both approaches to understand their impacts on robustness.

**Scaling LLMs.** As shown in Tab. 2, when scaling from Qwen2.5-VL-3B to Qwen2.5-VL-72B with a fixed vision encoder, overall $\text{Acc}_m$ increases and MVI-Sensitivity decreases. This trend suggests that stronger reasoning capacity can partially compensate for limited visual perceptual ability. However, the improvement is non-monotonic: within the mid-scale range (*e.g.*, from 7B to 32B), performance on misleading images degrades in certain categories, including *representation confusion* and *mirror reflection*.

**Enabling explicit reasoning via long-form CoT.** Next, we examine how explicit "long-thinking" influences model performance under misleading visual inputs. Unexpectedly, long-thinking variants of open-source models (*e.g.*, SAIL-VL2 and Qwen2-VL) exhibit lower $\text{Acc}_m$ on MVI-Bench, primarily due to declines in perception-heavy categories such as *visual resemblance* and *representation confusion*. This aligns with prior works (Tian et al., 2025; Liu et al., 2025a) showing that multimodal reasoning models' perceptual abilities weaken as reasoning processes. Analysis of paired instances (Fig. 5) reveals two failure patterns. First, reasoning becomes increasingly guided by historical thoughts rather than image content, a phenomenon termed "visual forgetting" (Tian et al., 2025) (More cases are provided in Appendix E.). Second, models increasingly over-attend to fine-grained details (*e.g.*, shape or color), which drives them away from correct answers. Interestingly, this behavior can be beneficial in categories such as *material*

*confusion* and *mirror reflection*, where models need to carefully examine and analyze visual details.

In contrast, proprietary models (like Gemini-2.5-Flash and GPT-5) show gains with explicit thinking, suggesting better training to effectively leverage extended reasoning without sacrificing perception. However, the thinking-enhanced Gemini-2.5-Flash still underperforms the non-reasoning GPT-5-Chat by 0.64% in $\text{Acc}_m$. This indicates that visual perception and reasoning are complementary in resisting misleading visual inputs, where visual perception serves as a fundamental prerequisite for extended reasoning in LVLMs.

### 4.4. Case Study: When LVLMs Outperform on Misleading Visual Inputs (RQ3)

While most models perform worse on misleading images, we observe a small but revealing set of counterintuitive instances (about 4% of total cases) where models answer misleading images correctly but fail on normal ones. To investigate, we employ an *attention-guided masking* approach inspired by recent interpretability studies (Jiang et al., 2025c; Stan et al., 2024; Zhu et al., 2025b; Kang et al., 2025; Tang et al., 2025): we visualize attention maps, identify high-attention regions, and mask them to test whether the model's answer changes (see Appendix D.4 for details).

**Findings.** Our analysis reveals that these cases arise from *spurious correlations* between visual artifacts and target labels. Fig. 6 shows a representative example. In the misleading image (Fig. 6 (b)), Qwen2.5-VL-7B misidentifies overlapping books as one and treats a receipt as another book, accidentally producing the correct answer ("2 books"). Masking the receipt (Fig. 6 (c)) immediately flips the prediction, confirming reliance on this false cue. This also explains failure on the normal image (Fig. 6 (a)): the model predicts "2" despite only one book present.

We speculate that this phenomenon stems from the lack of fine-grained supervision in current VQA training paradigms,

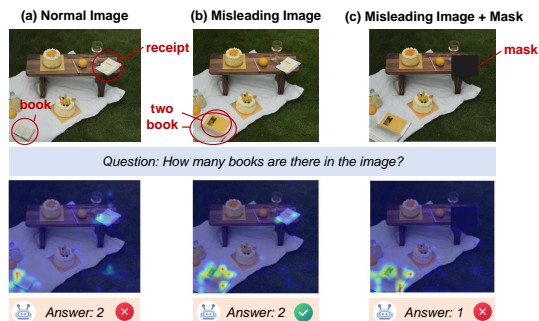

*Figure 6.* **Attention-guided masking for a counterintuitive instance.** Qwen2.5-VL-7B spuriously associates a receipt with a book. (a) On the normal image with one book, it answers incorrectly. (b) On the misleading image, it coincidentally answers "2" by counting the receipt as an extra book. (c) Masking the receipt flips the prediction, confirming the spurious correlation.

where models are supervised only on final answers without explicit rationale guidance. This weak supervision encourages models to exploit shortcut cues rather than learning true causal associations. Our findings also suggest that future VQA evaluation should consider not only answer correctness but also the underlying reasoning process.

## 5. Conclusion

This paper introduces MVI-Bench, the first comprehensive benchmark specifically designed to evaluate how misleading visual inputs undermine the robustness of LVLMs. In addition, we propose MVI-Sensitivity, a metric that characterizes LVLM robustness at a fine-grained level. Our evaluation reveals that existing LVLMs exhibit pronounced vulnerabilities to misleading visual inputs, and in-depth analysis on MVI-Bench provides actionable insights for developing more reliable and robust LVLMs.

## Acknowledgements

We thank Yuxin Wang and Yuhua Xie for their contributions to image collection and annotation of MVI-Bench.

## Impact Statement

This paper presents MVI-Bench, a benchmark for evaluating the robustness of Large Vision-Language Models (LVLMs) against misleading visual inputs. By examining model behavior from visual perception and reasoning perspectives, our work provides actionable insights for developing more reliable LVLMs. The analysis of counterintuitive cases further offers guidance for future training and evaluation methodologies. In developing MVI-Bench, ethical considerations were carefully addressed. We strictly complied with copyright regulations, collecting data only from publicly accessible sources. Generated images underwent manual review to mitigate potential biases. We believe this work contributes positively to responsible AI development and foresee no direct negative societal consequences.

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

# A. Limitations and Future Work

Although spurious-correlation cases constitute only a very small fraction of our benchmark, their implications may be more substantial in real-world settings. Modern LVLMs are trained on extremely large-scale datasets, often containing millions to billions of image–text pairs and even small proportions of these examples can accumulate into systematic biases. However, our analysis of spurious correlations relies fundamentally on the paired design of MVI-Bench, which enables counterfactual comparisons between normal and misleading images. As such, the methodology does not directly generalize to other datasets without further adaptations. Furthermore, our findings underscore broader challenges in current LVLM training and evaluation pipelines: shortcut learning remains difficult to mitigate under weak supervision that provides only answer labels without rationales, and existing VQA evaluations emphasize answer correctness without assessing whether predictions are causally grounded in visual evidence. Future work should explore training objectives and evaluation protocols that discourage shortcut exploitation and promote faithful, causally aligned visual reasoning.

# B. Ethics Statement

During the collection of major instances, we strictly complied with the copyright and licensing regulations of each social media platform, ensuring that data was collected only from publicly accessible posts and that no images were downloaded from sources explicitly prohibiting data reuse or redistribution. A portion of the dataset was additionally generated using Seedream, a powerful image generation model. While this model produces high-quality outputs, the generated content inevitably reflects the biases and limitations inherent in its training data. We recognize the ethical concerns associated with such models, including the potential to reinforce stereotypes or generate inappropriate content.

To minimize these risks, we applied careful dataset curation and conducted manual reviews throughout the construction process. We encourage future research to explore stronger methods for identifying and mitigating such biases, ensuring that both collected and generated content align with ethical standards and societal norms.

# C. More Details about MVI-Bench

### C.1. Data Curation Pipeline

The data curation pipeline is illustrated in Fig. 7.

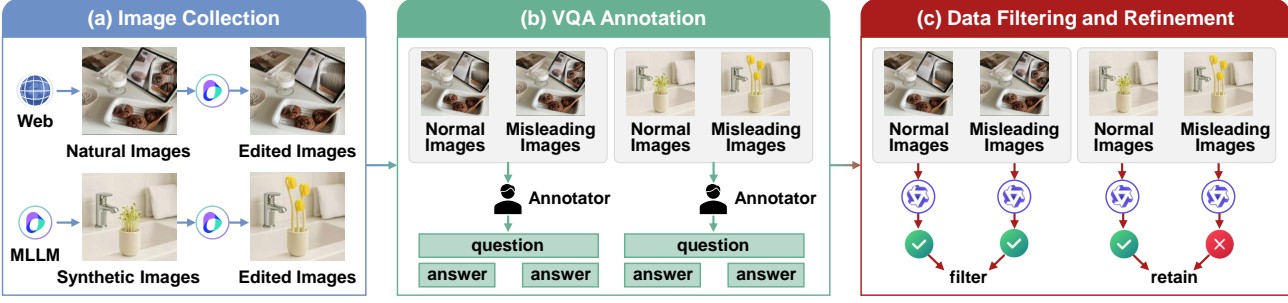

*Figure 7.* **Benchmark Curation Pipeline.** The pipeline starts with image collection, followed by VQA annotation, data filtering, and ultimately results in MVI-Bench. To ensure data quality, human verification is performed at each key stage to eliminate low-quality data, annotations, and ambiguous evaluation questions.

### C.2. Annotation Quality Control

This section provides additional details on the annotation process and quality control measures outlined in Section 3.

**Annotator qualifications.** As mentioned in Section 3, all annotators are trained human experts with at least a bachelor's degree. Before annotation, each annotator received detailed guidelines with category definitions, examples of valid/invalid instances, and common pitfalls to avoid.

**Verification process.** The annotation process involved multiple verification stages:

- *Initial annotation:* Three trained experts independently curated image pairs and crafted MCQs following the taxonomy.

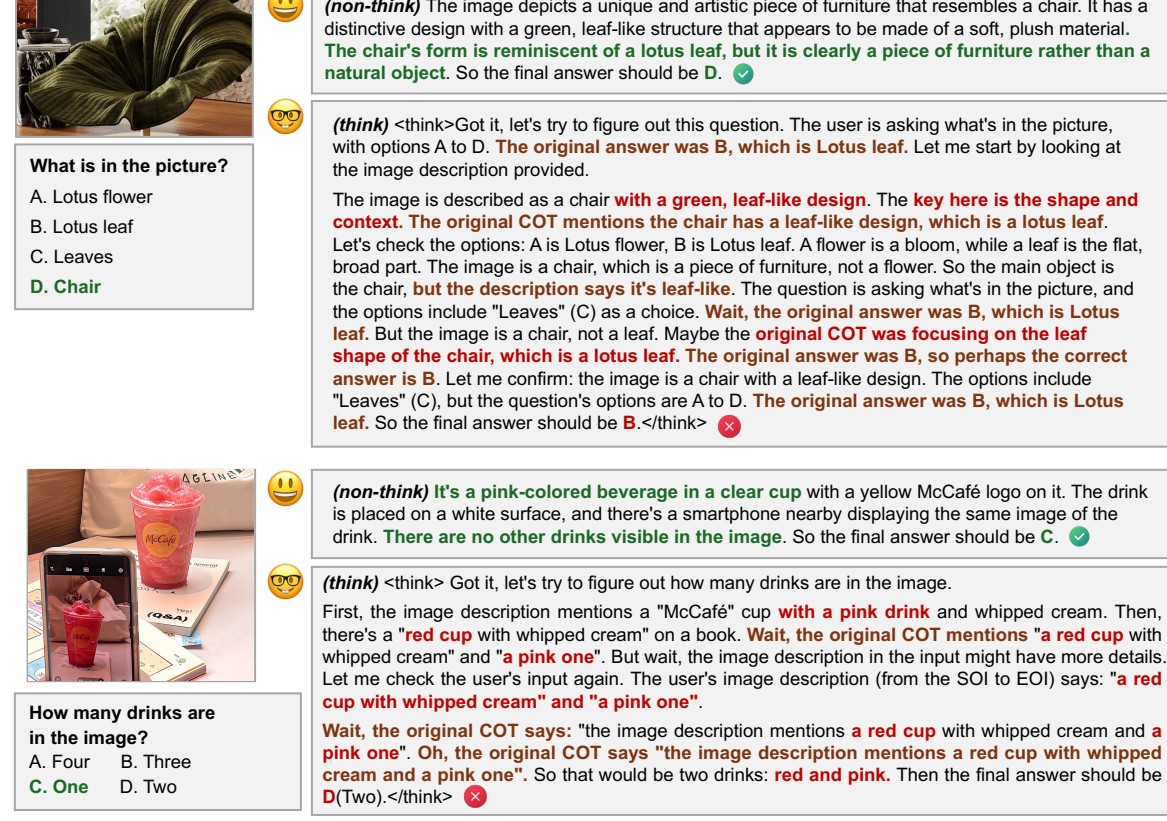

*Figure 8.* **Comparison between the "non-think" and "think" modes of SAIL-VL.** In the non-think mode, the model answers directly based on visual evidence, while in the think mode, the model is guided by **historical thoughts** and tend to **overemphasize fine details**.

- *Cross-review:* Each annotated VQA instance was then independently reviewed by additional experts to verify: (1) the image pair correctly reflects the intended misleading category, (2) the question is unambiguous, and (3) the ground-truth answer is correct.
- *Final check:* A final quality review was conducted by the first author to filter out any remaining ambiguous instances.

**Inter-annotator agreement.** To quantify annotation reliability, we computed Fleiss' Kappa on a randomly sampled subset of 200 instances, yielding $\kappa = 0.93$, indicating near-perfect agreement (Landis & Koch, 1977). This confirms that the annotation criteria were well-defined and consistently applied.

**Disagreement resolution.** When reviewers disagreed on an instance (approximately 5% of cases), the instance was discussed among annotators. If consensus could not be reached, the instance was discarded, as noted in Section 3. This conservative approach prioritizes annotation reliability over dataset size.

### C.3. More Examples

Fig. 9 presents additional examples from each category of MVI-Bench.

## D. Experiment Details

### D.1. Prompt Format for Caption-Assisted Inference

We describe the prompt format used for caption-assisted inference. After prompting a strong auxiliary LVLM with the instruction "Please describe the image.", we obtain a detailed caption that compensates for the limitations of the inference model's vision encoder. We then concatenate this caption with the corresponding image and question as input to the inference model, as illustrated below.

```
<image> Here are some additional information which are text descriptions
based on the image to assist you for answering the later question:{Caption}
{Question} Answer with the letter from the given choices directly.
```

### D.2. Model Details

Tab. 5 presents the release time and model sources of LVLMs used in MVI-Bench.

### D.3. Human Performance Evaluation

To establish a human performance baseline, we conducted a controlled study with four participants: two high school students and two university students. This selection covers different educational backgrounds to verify that MVI-Bench requires no specialized knowledge and remains solvable for the general population.

**Procedure.** Each participant independently completed all 1,248 VQA instances (624 pairs across six categories) in a randomized order. To simulate rapid visual perception in real-world scenarios, we imposed a 5-second time limit per question. If a participant failed to respond within 5 seconds, the response was marked as incorrect. This constraint also aligns with the fast inference setting of LVLMs in practical applications. Prior to the formal evaluation, each participant completed a practice session of 10 instances (not included in MVI-Bench) to familiarize themselves with the interface and time constraint.

**Results.** The reported human performance in Tab. 2 represents the average accuracy across all four participants. Despite the time constraint, humans achieve 98.24% $Acc_m$ with an MVI-Sensitivity of only 1.63%, demonstrating strong resilience to misleading visual cues. This result confirms that the visual misleading scenarios in MVI-Bench, while challenging for LVLMs, remain easily resolvable for humans through rapid perceptual processing and contextual reasoning.

### D.4. Implementation of Attention Visualization

To visualize where the model attends when generating an answer, we implement the *relative answer-to-image attention*. Specifically, for a given image $x$ and question $q$, we first feed the multimodal input sequence into the LVLM and extract the cross-attention weights from the first generated answer token to all image tokens between the `<|vision_start|>` and `<|vision_end|>` markers in each layer. The attention matrices are averaged across all attention heads, yielding the *answer-to-image-token* attention $A_{st}(x, q)$.

We then normalize this attention by its counterpart obtained from a generic instruction $q' =$"Write a general description of the image.", resulting in the *relative attention*:

$$A_{\text{rel}}(x, q) = \frac{A_{st}(x, q)}{A_{st}(x, q')}.$$

This normalization removes the model's default visual bias and highlights the image regions whose attention increases specifically in response to the question.

## E. Additional Case Studies on Long-form CoT

More cases illustrating the comparison between the Non-think and Think modes are presented in Fig. 8.

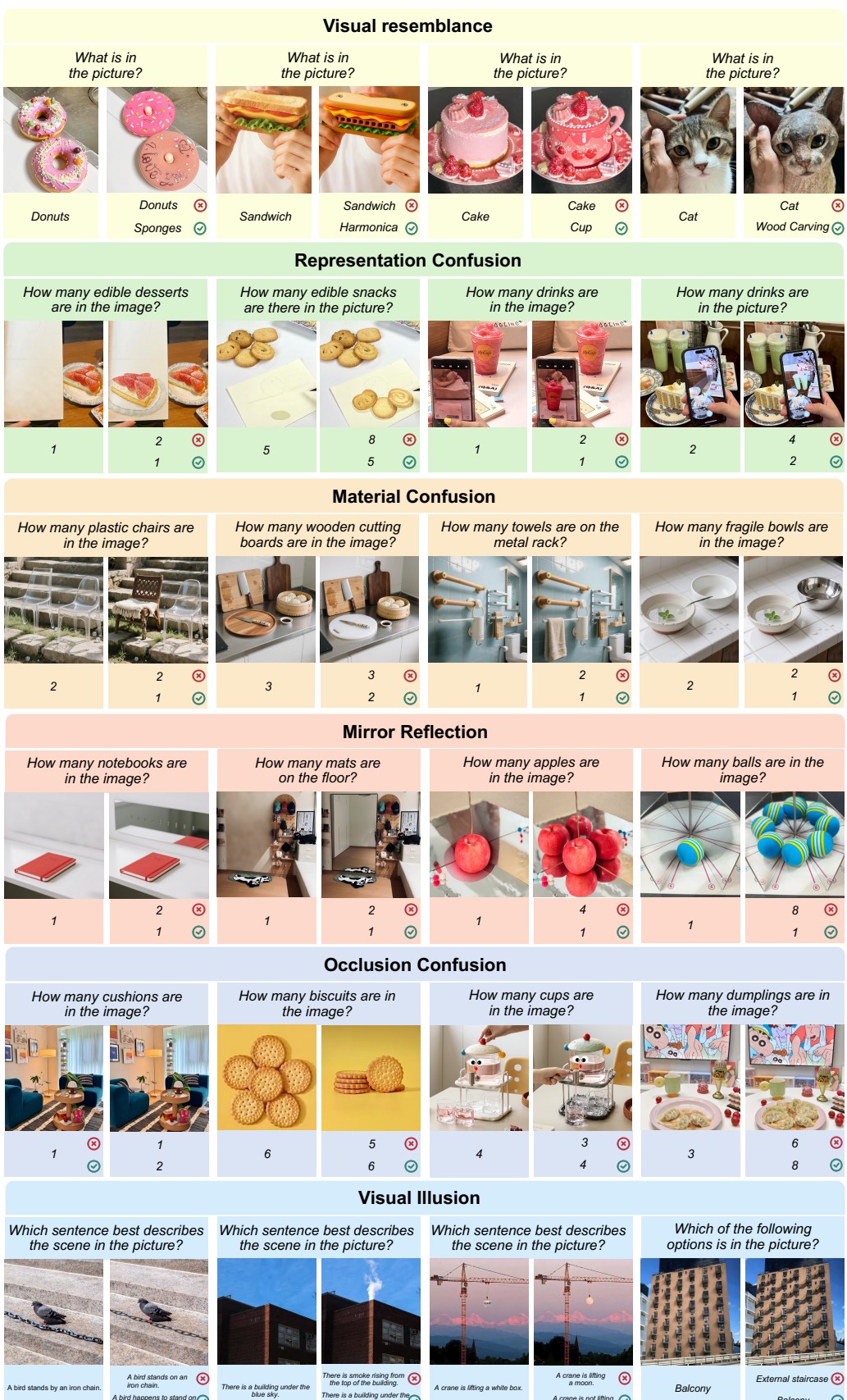

*Figure 9.* More Examples from six misleading categories defined in MVI-Bench.

*Table 5.* **The Release Time and Model Source of LVLMs Evaluated in MVI-Bench.**

| Model | Release Time | Source | URL |
|---|---|---|---|
| LLaVA-OneVision-7B (Li et al., 2024b) | 2024-08 | ByteDance | `https://github.com/LLaVA-VL/LLaVA-NeXT` |
| Molmo-7B-D-0924 (Deitke et al., 2025) | 2025-04 | allenai (Ai2) | `https://huggingface.co/allenai/Molmo-7B-D-0924` |
| SAIL-VL2-2B (Yin et al., 2025) | 2025-09 | ByteDance | `https://huggingface.co/BytedanceDouyinContent/SAIL-VL2-2B` |
| SAIL-VL2-8B (Yin et al., 2025) | 2025-09 | ByteDance | `https://huggingface.co/BytedanceDouyinContent/SAIL-VL2-8B` |
| InternVL3-2B (Zhu et al., 2025a) | 2025-08 | OpenGVLab | `https://huggingface.co/OpenGVLab/InternVL3-2B` |
| InternVL3-8B (Zhu et al., 2025a) | 2025-08 | OpenGVLab | `https://huggingface.co/OpenGVLab/InternVL3-8B` |
| InternVL3-78B (Zhu et al., 2025a) | 2025-08 | OpenGVLab | `https://huggingface.co/OpenGVLab/InternVL3-78B` |
| Qwen2-VL-72B-Instruct (Wang et al., 2024b) | 2024-09 | Alibaba | `https://huggingface.co/Qwen/Qwen2-VL-72B-Instruct` |
| Qwen2.5-VL-3B-Instruct (Bai et al., 2025) | 2025-01 | Alibaba | `https://huggingface.co/Qwen/Qwen2.5-VL-3B-Instruct` |
| Qwen2.5-VL-7B-Instruct (Bai et al., 2025) | 2025-01 | Alibaba | `https://huggingface.co/Qwen/Qwen2.5-VL-7B-Instruct` |
| Qwen2.5-VL-32B-Instruct (Bai et al., 2025) | 2025-01 | Alibaba | `https://huggingface.co/Qwen/Qwen2.5-VL-32B-Instruct` |
| Qwen2.5-VL-72B-Instruct (Bai et al., 2025) | 2025-01 | Alibaba | `https://huggingface.co/Qwen/Qwen2.5-VL-72B-Instruct` |
| Claude-3.7-Sonnet (Anthropic, 2025) | 2025-01 | Anthropic | `https://www.anthropic.com/news/claude-3-7-sonnet` |
| GPT-4o (Hurst et al., 2024) | 2024-05 | OpenAI | `https://platform.openai.com/` |
| GPT-4.1 (Achiam et al., 2023) | 2024-05 | OpenAI | `https://platform.openai.com/` |
| Gemini-2.5-Flash (Team et al., 2023) | 2025-06 | Google | `https://gemini.google.com/app` |
| Gemini-2.5-Pro (Team et al., 2023) | 2025-06 | Google | `https://gemini.google.com/app` |
| GPT-5 Chat (OpenAI, 2025) | 2025-08 | OpenAI | `https://platform.openai.com/` |

