# OpenReview forum: "MVI-Bench: A Comprehensive Benchmark for Evaluating Robustness to Misleading Visual Inputs in LVLMs"
_ICML.cc/2026/Conference — ICML 2026 regular_

### Official Review · Reviewer_omVH · 2026-03-12

**Soundness:** 2
**Presentation:** 3
**Significance:** 3
**Originality:** 2
**Overall Recommendation:** 4
**Confidence:** 4

**Summary:**

The authors propose **MVI-Bench**, a comprehensive benchmark designed to evaluate how misleading visual inputs undermine the robustness of **Large Vision-Language Models (LVLMs)**. They further introduce the **MVI-Sensitivity** metric to enable fine-grained robustness assessment. Extensive experiments are conducted on **18 LVLMs** to analyze their performance under misleading visual conditions.

**Compliance With Llm Reviewing Policy:**

Affirmed.

**Final Justification:**

My concerns have been fully addressed, and I have increased my score. The benchmark proposed by the authors helps evaluate the robustness of MLLMs.

**Key Questions For Authors:**

1.When Qwen uses GPT-4.1 as the captioning model, does the caption generated by GPT-4.1 already reveal the correct answer? If so, the Qwen model may not need to rely on visual perception and could instead answer the question through textual shortcuts. In this case, does the evaluation actually measure the capability of GPT-4.1 (overall 62.68) or that of the Qwen model?

**Limitations:**

yes

**Strengths And Weaknesses:**

Strengths:

1. The authors establish a taxonomy of misleading visual inputs and divide the evaluation into three dimensions: visual concepts, visual attributes, and visual relations.

2. Experiments are conducted on a diverse set of LVLMs.

Weaknesses:

1. The dataset contains only 624 pairs of VQA instances, which may be insufficient to rigorously support the experimental conclusions.

2. The design of the MVI-Sensitivity metric is relatively simple.

3. The overall idea of the paper is quite similar to IllusionBench. Although the paper provides a categorization into multiple types of misleading visual inputs, the amount of data in each category is still limited. What's more, according to Table 1, the images in IllusionBench are all natural images. Overall, the novelty of the paper appears to be limited.

---

> ### Author Rebuttal · Authors · 2026-03-30
>
> **To reviewer omVH.** Thank you for your thoughtful review and valuable suggestions.
>
> **W1: Dataset scale.** (1) **Paired design yields higher information density.** MVI-Bench's paired design makes each instance pair a controlled comparison, yielding higher information density than unpaired benchmarks. Compared to related works (Tab. 1), none of IllusionVQA (374 images), IllusionBench+ (1,051 images), or O-Bench (1,365 images, synthetic only) adopts a paired design. (2) **Higher per-instance annotation cost.** Each MVI-Bench pair requires constructing semantically matched normal and misleading images (with edited images manually crafted per instance), designing MCQs with misleading distractors, and multi-round cross-review, making the per-instance cost substantially higher. As described in Appendix C.2, our curation process explicitly prioritizes annotation reliability over dataset size. (3) **Sufficient to support stable conclusions.** Our conclusions are not drawn from isolated examples, but from consistent patterns across 18 LVLMs. We repeatedly observe substantial performance degradation from normal to misleading inputs across models, which suggests that the benchmark size is sufficient to reveal stable robustness trends.
>
> **W2: MVI-Sensitivity metric.** We believe its simplicity is in fact a design strength instead of limitation. （1）**Established precedent.** Many widely adopted metrics similarly embrace simplicity, such as Accuracy in VQA benchmarks, Attack Success Rate in adversarial robustness, and Flip Rate in consistency evaluation. （2）**Clear interpretability.** It directly quantifies the relative performance degradation from normal to misleading images, requiring no additional background knowledge. （3）**Design-grounded validity.** It is meaningful precisely because of our paired design, where each instance pair controls all variables except the misleading cue, so a simple relative change metric suffices to capture robustness differences.
>
> **W3: Novelty vs. IllusionBench.** We respectfully clarify the key differences along three dimensions. (1) **Different research focus**. MVI-Bench targets misleading visual scenarios that naturally arise in real-world interactions (e.g., representation confusion, mirror reflection). In contrast, IllusionBench+ primarily focuses on classical cognitive illusions (e.g., geometric distortions, Ishihara images), which are more abstract and artificially constructed. The two benchmarks address fundamentally different research directions.(2) **Different scope and taxonomy**. MVI-Bench provides a comprehensive taxonomy grounded in visual primitives, covering three hierarchical levels with six categories spanning from low-level recognition to high-level spatial reasoning. IllusionBench+ focuses solely on visual illusion. (3) **Fundamental design difference**. MVI-Bench's paired design with MVI-Sensitivity enables controlled isolation of misleading visual effects, supporting analyses such as counterintuitive case discovery (Sec. 4.4) , which IllusionBench+ cannot provide with absolute accuracy alone.
>
> **Key Question: textual shortcuts.** The caption is a generic image description, generated without access to the downstream question or answer options. As a result, it does not directly reveal the correct answer in a question-aware manner. However, such captions naturally contain coarse semantic information about the image content, which can provide supplementary context to the inference model.
> To directly examine this issue, we conduct a caption-only experiment: we provide only GPT-4.1's caption and the question (without the image) to Qwen2.5-VL-7B:
>
> **Table: Acc_m(Accuracy on misleading images, %) of different models under different settings.** Image-only: standard inference with image input. Image + Caption: image plus GPT-4.1-generated caption. Caption-only: only GPT-4.1-generated caption without image input.
> |  Model| Input| Overall |Res | Rep | Mat | Mir | Occ | Ill |
> |--|--|--|--|--|--|--|--|--|
> | Qwen2.5-VL-7B|Image-only (baseline) | 45.99 |53.33 | 60.75 | 46.60 | 43.72 | 29.52 | 42.00 |
> | Qwen2.5-VL-7B|Image + Caption|  53.85 |58.10 | 71.96 | 49.51 | 53.85 | 33.33 | 56.00 |
> | GPT-4.1| Image-only| 62.82 |73.33 | 74.77 | 55.00 | 67.31 | 49.52 | 58.00 |
> | Qwen2.5-VL-7B|Caption-only | 37.66 |52.38 | 53.27 | 33.01 | 33.65 | 12.38 | 41.00 |
>
> Caption-only achieves 37.66% overall accuracy, which is clearly below both image-only (45.99%) and image+caption (53.85%). This suggests that the caption alone does not fully reveal the answer or substitute for direct visual input. At the same time, its non-trivial performance confirms that the caption does provide useful coarse semantic information.

---

> > ### Author Rebuttal · Reviewer_omVH · 2026-04-02
> >
> > Some of my concerns have been addressed. However, I still believe that, as a benchmark, ensuring sufficient scale is necessary to guarantee quality, even though data collection may involve practical costs or challenges. The current dataset size in the paper appears relatively limited. Therefore, I will keep my score unchanged. If the authors can provide more data samples and a more comprehensive analysis of data diversity, I would be willing to adjust my score accordingly.

---

> > > ### Author Response · Authors · 2026-04-02
> > >
> > > Thank you for your follow-up.  We provide a more comprehensive diversity analysis and further clarification on dataset scale below.
> > >
> > > **1. More Detailed Diversity Analysis.**
> > >
> > > (1) **Object diversity.** As shown in Fig. 3(c), the objects in MVI-Bench images cover 6 major semantic domains spanning 30+ fine-grained subcategories:
> > >
> > > | Domain | # | Representative subcategories |
> > > |---|---|---|
> > > | Furniture | 453 | Table (160), Chair (71), Lighting (65), Bed (51), Cabinet (39), Cushion (30) |
> > > | Decorations | 352 | Potted plant (114), Mirror (110), Painting (41), Vase (38), Stickers (30) |
> > > | Food | 201 | Fruit (58), Snack (34), Cookie (22), Sushi (17), Cakes (15), Ice Cream (14) |
> > > | Others | 170 | Clothing (60), Person (55), Pet (20), Utensils (19), Books (16) |
> > > | Nature | 164 | Tree (52), Flower (38), Sky (23), Cloud (17), Leaf (14), Grass (10) |
> > > | Toy | 140 | Figures (31), Building sets (23), Toy vehicles (16), Balls (11), Plush toy (9) |
> > >
> > > (2) **Question type diversity.** MVI-Bench covers five question types: how-many, what, which, where, and others.
> > > As detailed in our response to Reviewer NbrN (W1), we intentionally exclude is/are-type questions to prevent known textual bias [1] from confounding visual misleading evaluation, and exclude why-type questions as they depend more on prior knowledge than visual evidence. Among the remaining formats, how-type (counting) questions dominate because they most directly expose misleading effects in four categories: reflections cause overcounting (Mir.), occlusions cause undercounting (Occ.), and qualifying adjectives in counting naturally test material discrimination (Mat.) and real-vs-depicted distinction (Rep.). The what-type and which-type questions serve the remaining two categories: identification for Visual Resemblance and scene description for Visual Illusion.
> > >
> > > [1] "Unveiling the Ignorance of MLLMs: Seeing Clearly, Answering Incorrectly". (CVPR2025)
> > >
> > > (3) **Image source diversity.** Beyond the three source types reported in Fig. 3(b) (natural 52.32%, edited 38.06%, synthetic 9.62%), the natural images were collected from multiple social  media platforms: rednote (38%), Instagram (28%), X (24%), and Weibo (10%), ensuring broad geographic and cultural coverage.
> > >
> > >
> > > **2. On dataset scale.**
> > >
> > > Thank you for your suggestion. The primary goal of MVI-Bench is to investigate whether current LVLMs are vulnerable to misleading visual inputs that humans can easily resolve. **At the current scale (1,248 instances), consistent performance degradation patterns across all 18 evaluated LVLMs already validate this finding**.
> > >
> > > To further verify the stability of our findings, we evaluate Qwen2.5-VL-7B on random subsets of varying sizes, where each subset uniformly samples 10, 20, 40, 60, and 80 pairs per category respectively. The overall results are as follows:
> > >
> > > | Count | Acc_n (%) | Acc_m (%) | Sens (%) |
> > > |---|---|---|---|
> > > | 120 (10×2×6)| 84.42 | 49.40 | 41.48 |
> > > | 240 | 87.49 | 49.37 | 43.57 |
> > > | 480 | 85.00 | 48.30 | 43.17 |
> > > | 720 | 84.42 | 47.38 | 43.95 |
> > > | 960 | 83.13 | 46.39 | 45.40 |
> > > | 1,248 (full) | 81.89 | 45.99 | 43.84 |
> > >
> > > Across all scales, Acc_m is consistently and substantially lower than Acc_n, and Sens remains stable around 43%, **confirming that our core finding holds regardless of dataset size.**
> > >
> > > Finally, we would like to emphasize that MVI-Bench, **with its 1,248 carefully curated samples, is comparable in size to several widely used and high-quality VQA benchmarks**, such as MM-Star [2] (1,500 instances), MM-Vet [3] (218 instances), MMSI-Bench [4] (1,000 instances) and LLaVA-in-the-Wild [5] (60 instances). While we agree that a larger dataset scale would always be beneficial, we believe that the current size of MVI-Bench is adequate and its evaluation results are trustworthy.
> > >
> > > [2] "Are we on the right way for evaluating large vision-language models?." (Neurips 2024)
> > >
> > > [3] "MM-Vet: Evaluating Large Multimodal Models for Integrated Capabilities." (ICML2024)
> > >
> > > [4] "MMSI-Bench: A Benchmark for Multi-Image Spatial Intelligence." (ICLR2026)
> > >
> > > [5] "Visual instruction tuning."  (NeurIPS 2023)
> > >
> > > We hope the comprehensive diversity analysis and dataset scale clarification above address your remaining concerns.

---

### Official Review · Reviewer_NbrN · 2026-03-13

**Soundness:** 3
**Presentation:** 4
**Significance:** 3
**Originality:** 3
**Overall Recommendation:** 5
**Confidence:** 4

**Summary:**

This paper proposes a new benchmark, MVI-Bench, which consists of visually misleading images across three dimensions: (1) visual concept, (2) visual attribute, and (3) visual relationship. They show that existing LVLMs are vulnerable to these small and fine-grained changes to the image.

**Compliance With Llm Reviewing Policy:**

Affirmed.

**Final Justification:**

The paper has a clear contribution toward the robustness of LVLMs and showcases a scenario where LVLMs are misled by the visual details that are quite simple for humans to do.

**Key Questions For Authors:**

See weakness

**Limitations:**

yes

**Strengths And Weaknesses:**

## Strength
* The goal of the benchmark is clear and well-defined
* The data quality looks good from the examples provided in the paper
* The evaluation is comprehensive, and the analysis on counter-intuitive samples is reasonable.

## Weakness
* While the object coverage is good, it's unclear whether the text query is mostly counting based, as the examples shown in the paper seem to suggest that the majority of the queries are counting-based.
* The existing analysis provides some insights on visual reasoning and perception capability. However, it'll be interesting to see whether the model is more robust if you prompt the LVLM to pay attention to the edit (for instance, in the spoon conductivity case, ask the model to pay attention to the material of the spoons). If not, then it might suggests a fundamental limitation in the visual encoder rather than the attention being misled.

---

> ### Author Rebuttal · Authors · 2026-03-30
>
> **To reviewer NbrN.** Thank you for your thoughtful review and valuable suggestions.
>
> **W1: Text Queries.** While counting queries are common across several categories, our benchmark also includes identification queries (Visual Resemblance) and scene description queries (Visual Illusion). We would like to clarify the rationale behind our query design, which follows two principles:
> (1) **Eliminating textual misleading confounds**. Prior work [1] reveals that LVLMs exhibit inherent bias toward is/are-type questions, often answering incorrectly even with correct visual perception. **Since MVI-Bench aims to isolate the effect of visual misleading, we deliberately avoid such formats to prevent textual bias from confounding our evaluation.** We also exclude why-type questions, as they are less tightly grounded in visual evidence and more likely to depend on prior knowledge. As a result, our queries primarily adopt what-type, how-type, and which-type formats.
>  (2) **Matching query format to each category.** We choose the query format that most directly exposes the misleading effect in each category.  For Mirror/Occlusion, counting most directly exposes the misleading effects (reflections cause overcounting; occlusions cause undercounting). For Material/Representation, counting with qualifying adjectives naturally tests material discrimination and real-vs-depicted object distinction. Visual Resemblance naturally suits identification queries, and Visual Illusion suits scene description queries.
>
> [1] Unveiling the Ignorance of MLLMs: Seeing Clearly, Answering Incorrectly (Liu et al., 2025b)
>
> **W2: Prompt the LVLM.** We further investigate whether explicitly prompting the LVLM to attend to misleading cues improves robustness. To this end, we prepend simple category-specific hints to the original question:
>
> - Visual Resemblance: "Look carefully at what the object actually is."
> - Representation Confusion: "Distinguish between real objects and 2D representations."
> - Material Confusion: "Pay attention to the material of the objects."
> - Mirror Reflection: "Note that some objects may be reflections in mirrors."
> - Occlusion Confusion: "Some objects may be partially occluded."
> - Visual Illusion: "Be aware of potential visual illusions."
>
> **Table: Acc_m (Accuracy on misleading images, %) of Qwen2.5-VL-7B with and without category-specific hints.**
> | |Overall | Res. | Rep. | Mat. | Mir. | Occ. | Ill. |
> |--|--|--|--|--|--|--|--|
> | baseline| **45.99** |53.33 | 60.75 | 46.60 | 43.72 | 29.52 | 42.00 |
> | Category-specific hint | **48.56** |54.28 | 62.62 | 48.54 | 44.23 | 29.52 | 52.00 |
>
> As shown above, with explicit category-specific hints, overall Acc_m improves only marginally. This suggests that the challenge cannot be fully addressed by such guidance. Moreover, this setting is impractical in real-world scenarios, as LVLM-based systems typically do not know in advance which type of misleading visual cue is present. We will add this analysis to the revision.

---

> > ### Author Rebuttal · Reviewer_NbrN · 2026-04-03
> >
> > My concerns are fully resolved by the rebuttal. I raise my score by 1. It'll be very interesting to see how the community can improve LVLMs on the misleading input.

---

### Official Review · Reviewer_GQLd · 2026-03-16

**Soundness:** 3
**Presentation:** 3
**Significance:** 2
**Originality:** 3
**Overall Recommendation:** 4
**Confidence:** 4

**Summary:**

This paper introduces MVI-Bench, a benchmark for evaluating the robustness of large vision-language models under misleading visual inputs. The benchmark is organized around six categories of misleading visual phenomena and uses a paired setup: each question is evaluated on a normal image and a misleading counterpart while keeping the question and answer options fixed. The paper evaluates a broad set of open- and closed-source LVLMs. The main empirical finding is that even strong models remain quite vulnerable to misleading visual cues, whereas humans are largely unaffected.

**Compliance With Llm Reviewing Policy:**

Affirmed.

**Key Questions For Authors:**

Where the real images specifically come from and how they were collected？

**Limitations:**

Yes

**Strengths And Weaknesses:**

**Strengths**

1. The paper targets a meaningful and under-measured failure mode. Robustness to misleading visual cues is not the same thing as robustness to misleading prompts, generic hallucination, or adversarial perturbations. The paired construction makes the evaluation relatively clean: if the question and options are held fixed and only the image changes, then the observed drop is much easier to attribute to the image-side misleading cue.

2. The main empirical result is convincing at the descriptive level.
Across a fairly broad set of models, performance drops substantially on misleading images, while human performance stays near ceiling. I think this supports the paper’s central descriptive claim that current LVLMs are still brittle under these visual traps.

**Weaknesses**

1. The paper does not clearly disentangle whether the model fails because it did not perceive the image correctly, or because it recognized the visual content but misinterpreted its status within the scene. For example, the model may treat a person in a mirror as an additional real entity in the physical scene. This type of failure is not purely about perception; it also involves scene understanding and spatial reasoning.

2. The paper uses caption-assisted inference to support the claim that perception is the key factor, but I do not think this evidence is clean enough. A caption provides not only stronger visual perception, but also additional semantic abstraction and language-side guidance.

3. The evaluation is missing more recent models. By the submission date, newer frontier models such as Gemini 3.0 Pro, Qwen3-VL, and Claude Sonnet 4.5 had already been released and should have been considered.

---

> ### Author Rebuttal · Authors · 2026-03-30
>
> **To reviewer GQLd.** Thank you for your thoughtful review and valuable suggestions.
>
> **W1: Disentangling failures.** We agree that certain categories in MVI-Bench, such as Mirror Reflection, involve not only visual perception but also scene understanding and spatial reasoning. This is consistent with our taxonomy: we categorize mirror reflection under Visual Relationship, rather than Visual Concept, precisely because resolving such cases requires reasoning about higher-order spatial arrangements (Sec. 3.1).
>
> At the same time, our intention was not to attribute all failure modes in MVI-Bench purely to perception. In fact, our analysis in Sec. 4.3 shows that scaling the LLM backbone can also improve overall performance on MVI-Bench, although the gains are not always stable across categories. This suggests that higher-level reasoning also plays an important role.
>
> More precisely, perception and reasoning are complementary: reliable visual perception provides the necessary foundation for subsequent reasoning (e.g., recognizing reflected content before determining whether it corresponds to a real or virtual object). We will revise Sec. 4.3 to more clearly convey this relationship.
>
>
> **W2:Caption-assisted evidence.** We agree that captions are not a purely perceptual signal and may carry additional semantic abstraction. To further investigate this, we conducted a **caption-only** experiment, where only the GPT-4.1-generated caption and the question are provided to the inference model (Qwen2.5-VL-7B), without any image input.
>
> **Table: Acc_m(Accuracy on misleading images, %) of Qwen2.5-VL-7B under different settings.** Image-only: standard inference with image input. Image + Caption (GPT4.1): image plus GPT-4.1-generated caption. Caption-only: only GPT-4.1-generated caption without image input.
> | Setting | Overall |Res | Rep | Mat | Mir | Occ | Ill |
> |--|--|--|--|--|--|--|--|
> | Image-only (baseline) | 45.99 |53.33 | 60.75 | 46.60 | 43.72 | 29.52 | 42.00 |
> | Image + Caption (GPT-4.1) |  53.85 |58.10 | 71.96 | 49.51 | 53.85 | 33.33 | 56.00 |
> | Caption-only | 37.66 |52.38 | 53.27 | 33.01 | 33.65 | 12.38 | 41.00 |
>
> As shown above, caption-only achieves 37.66% overall accuracy, which remains clearly below both image-only (45.99%) and image+caption (53.85%). Notably, its accuracy on Occlusion Confusion drops to just 12.38%, well below random chance (25%), suggesting that captions mainly preserve coarse semantic information but fail to capture the fine-grained details needed to resolve such cases. Therefore, captions cannot fully substitute for raw visual input in misleading visual scenarios.
>
> We will revise the manuscript to present caption-assisted inference as supportive evidence that improving perception-related information can improve robustness under misleading visual inputs.
>
> **W3: More models.** We evaluate several recent LVLMs on MVI-Bench. As shown in table below, even the strongest newly added model, Gemini-3-Pro, achieves an overall Acc_m of 68.59% with an MVI-Sensitivity of 21.76%, which still remains far below human performance (Acc_m = 98.24%, Sens = 1.63%).
>
> **Table: Performance (%)  comparison on MVI-Bench.** Acc_n and Acc_m denote accuracy on normal and misleading images, respectively. Sens denotes MVI-Sensitivity, measuring the relative performance degradation from normal to misleading inputs.
>
> | | **Overall** | | Res. | | | Rep. | | | Mat.| | | Mir. | | | Occ. | | | Ill. | | |
> |--|--|--|--|--|--|--|--|--|--|--|--|--|--|--|--|--|--|--|--|--|
> | Model | **Acc_m↑** | **Sens↓** | Acc_n | Acc_m↑ | Sens↓ | Acc_n | Acc_m↑ | Sens↓ | Acc_n | Acc_m↑ | Sens↓ | Acc_n | Acc_m↑ | Sens↓ | Acc_n | Acc_m↑ | Sens↓ | Acc_n | Acc_m↑ | Sens↓ |
> | Gemini-3-Pro | **68.59** | **21.76** | 92.38 | 81.90 | 11.34 | 81.31 | 73.83 |9.20 | 88.35 | 63.11| 28.57 | 95.19| 67.31 | 29.29 | 79.05 | 50.48 | 36.14 | 90.00 | 75.00| 16.67 |
> | Qwen3-VL-7B | **52.56** | **40.58** | 97.14 | 68.57 | 29.41 | 83.18 | 51.40 | 38.21 | 83.50 | 52.43 | 37.21 | 93.27 | 42.31 | 54.64 | 82.86 | 43.81 | 47.13 | 91.00 | 57.00 | 37.36 |
> | Claude Sonnet 4.5 | **47.92** | **34.43** | 88.57 | 55.24 | 37.63 | 71.03 | 60.75 | 14.47 | 56.31 | 38.83 | 31.03 | 76.92 | 29.81 | 61.25 | 57.14 | 42.86 | 25.00| 89.00 | 60.00 | 32.58 |
>
>
> **Key question: Sources of real images**:  As described in our Ethics Statement (Appendix B), we complied with the copyright and licensing regulations of each platform during data collection. Specifically, the natural images in MVI-Bench were manually collected by three trained annotators from multiple social media platforms, including rednote, Instagram, X, and Weibo. This multi-platform collection strategy was intended to improve diversity in scene types, object categories, and cultural contexts. The collection process was guided by our taxonomy: for each misleading category, annotators searched for candidate images using category-relevant keywords and selected samples that satisfied the corresponding category definition.

---

> > ### Author Rebuttal · Reviewer_GQLd · 2026-04-03
> >
> > The rebuttal is helpful, but I still think this concern is only partially resolved.
> >
> > My main issue is that the paper still does not clearly separate failures of perception from failures where the model recognizes the visual content but misinterprets its status in the scene, so I remain somewhat unconvinced by the stronger claim that perception is the key bottleneck.
> >
> > The added caption-only result is useful and does strengthen the discussion. Still, I do not think it fully isolates perception, since the caption may also provide higher-level semantic disambiguation rather than just better visual information.

---

> > > ### Author Response · Authors · 2026-04-05
> > >
> > > Thank you for the thoughtful follow-up. Your comment helps us better calibrate this aspect of our claims.
> > >
> > > **1. Regarding disentangling perception from scene understanding.**
> > >
> > > We agree that our current analysis does not cleanly separate failures of perception from failures of scene understanding. Fully disentangling the two remains challenging, and we attempt to investigate this from an alternative angle: whether explicitly enhancing scene understanding via textual prompts can improve robustness under misleading visual conditions. Specifically, for each category, we prepend a scene-understanding prompt that describes the relevant scene context to the original question:
> > >
> > > - Visual Resemblance: "The scene contains objects that are visually similar but semantically different from each other."
> > > - Representation Confusion: "The scene includes both real physical objects and their two-dimensional depictions such as photographs or paintings."
> > > - Material Confusion: "The scene contains objects whose materials may look similar at a coarse level but differ fundamentally."
> > > - Mirror Reflection: "The scene includes both real physical objects and their virtual reflections in a mirror."
> > > - Occlusion Confusion: "Some objects in the scene are partially occluded by other objects, so their full extent may not be visible."
> > > - Visual Illusion: "The spatial arrangement in the scene may create visual impressions that differ from the actual physical layout."
> > >
> > >
> > > **Table: Acc_m (Accuracy on misleading images, %) of Qwen2.5-VL-7B with and without  scene-understanding prompts.**
> > > | |Overall | Res. | Rep. | Mat. | Mir. | Occ. | Ill. |
> > > |--|--|--|--|--|--|--|--|
> > > | baseline| **45.99** |53.33 | 60.75 | 46.60 | 43.72 | 29.52 | 42.00 |
> > > | Scene-understanding prompt | **48.24** |56.19 | 63.55 | 48.54 | 46.15 | 31.43 | 43.00 |
> > >
> > > As shown above, when the model is explicitly guided to understand the scene structure, the improvement is marginal (+2.25%).
> > >
> > > Accordingly, we will revise the original claim in the paper. Our original claim ("visual perception remains the primary bottleneck in resisting misleading visual inputs and serves as a fundamental prerequisite for robust extended reasoning in LVLMs") is indeed stronger than intended. We will adopt a more precise and cautious characterization: **visual perception and reasoning are complementary, where the former serves as a prerequisite for the latter.**
> > >
> > >
> > > **2. Regarding caption-assisted evidence.**
> > >
> > > We agree with your point that captions may provide higher-level semantic disambiguation beyond better visual information. In our original paper, we intended to present caption-assisted inference as “a proxy for enhanced perception” rather than as a direct isolation of perception. However, we acknowledge that this framing was not sufficiently clear and may have led to a stronger reading than intended. **We will ensure the revised manuscript presents caption-assisted inference as supportive evidence rather than a clean isolation of perception.**
> > >
> > > More broadly, MVI-Bench is built on a carefully paired design where each image pair shares nearly identical semantic content and identical textual input, differing only in subtle misleading visual cues. Its primary goal is to investigate whether current LVLMs are vulnerable to misleading visual inputs that humans can easily resolve. The analyses in Sec. 4.3 are intended as exploratory investigations into the factors behind the observed vulnerability, rather than to establish a definitive causal attribution between perception and reasoning.
> > >
> > > We thank you again for highlighting these distinctions, and we will refine the relevant  claims in the revised version.

---

### Official Review · Reviewer_EG62 · 2026-03-16

**Soundness:** 3
**Presentation:** 3
**Significance:** 3
**Originality:** 3
**Overall Recommendation:** 5
**Confidence:** 4

**Summary:**

This paper presents MVI-Bench, a misleading VQA benchmark for testing the robustness of current LVLMs.

The benchmark is well designed, with reasonable types of misrepresentation, including Visual Concept, Visual Attribute, and Visual Relationship, and is curated carefully through small-model filtering. The benchmark consists of ~1250 samples, covering a wide range of image types and objects.

Benchmarking results show interesting findings, such as the perception bottleneck, scaling dimensions (LLM backbones or CoT reasoning), and provide valuble insights for future improvements.

**Compliance With Llm Reviewing Policy:**

Affirmed.

**Final Justification:**

The rebuttal with updated results fully addressed my concerns. I keep my original recommendations.

**Key Questions For Authors:**

- What's the performance of a strong text-only LLM and the accuracy without image inputs? This would be a useful sanity check for the language bias.

**Limitations:**

yes

**Strengths And Weaknesses:**

## Pros

- The visual misleading benchmark test is an important dimension for current LVLMs, revealing potential robustness issues of current LVLMs.
- The dataset is well curated with careful quality test (small models filtering and human evaluation).
- The benchmark results are interesting, showcasing the importance of visual perception and fine-grained supervision.


## Cons
- The current formats still rely on MCQs, including an open-ended subset, and examining the option difficulty would be better.
- (Minor) As LVLMs are evolving rapidly, it would be better if the authors could test more recent LVLMs, especially reasoning-enhanced LVLMs such as K2.5, MiMo-VL, and Gemini 3 series.

---

> ### Author Rebuttal · Authors · 2026-03-30
>
> **To reviewer EG62.** Thank you for your thoughtful review and valuable suggestions.
>
> **Cons1: Choice of MCQs Format.** We adopt MCQs for three reasons.
> (1) **Evaluation Reliability.** Following well-established benchmarks (MMMU, MMBench, SEED-Bench), we adopt MCQs to enable straightforward answer extraction and deterministic evaluation, thereby avoiding the additional noise introduced by LLM-as-a-judge in open-ended evaluation. (2) **Experimental Control.** MVI-Bench is designed to measure LVLM robustness through the relative performance drop between normal and misleading images, where each pair shares the same text input and differs only in the visual content. Open-ended responses can introduce additional confounding factors, such as prior-knowledge bias, making it harder to isolate the effect of misleading visual cues. By constraining the output space, MCQs provide a more controlled and fair comparison.(3) **Focus of the Benchmark.** Under MVI-Bench's paired design, our goal is to directly and effectively reveal the impact of misleading visual cues, rather than evaluating the quality of open-ended versus MCQ responses.
>
>
> **Cons1: Option Difficulty.** MVI-Bench covers two types of questions: counting-based (Representation, Material, Mirror, Occlusion) and non-counting (Visual Resemblance, Visual Illusion).
>
> **For counting-based categories**, we construct challenging distractors by sampling nearby numerical values around the ground-truth count. For example, if the correct answer is n=2, the distractors are chosen from values close to n, such as {0, 1, 3}. This prevents models from ruling out obviously incorrect options and encourages precise visual counting.
>
> **For non-counting categories**, as described in Sec. 3, annotators are required to include at least one distractor that appears plausible under the misleading visual cues. For example, when an image depicts a stool resembling a mushroom, the correct answer is “stool,” while “mushroom” is explicitly included as a distractor. This ensures that the options are meaningfully challenging rather than trivially dismissible.
>
>
> **Cons2: More models.**  We evaluate several recent LVLMs on MVI-Bench. The results are summarized in table below:
>
> **Table: Performance (%)  comparison on MVI-Bench.** Acc_n and Acc_m denote accuracy on normal and misleading images, respectively. Sens denotes MVI-Sensitivity, measuring the relative performance degradation from normal to misleading inputs.
>
> | | **Overall** | | Res. | | | Rep. | | | Mat.| | | Mir. | | | Occ. | | | Ill. | | |
> |--|--|--|--|--|--|--|--|--|--|--|--|--|--|--|--|--|--|--|--|--|
> | Model | **Acc_m↑** | **Sens↓** | Acc_n | Acc_m↑ | Sens↓ | Acc_n | Acc_m↑ | Sens↓ | Acc_n | Acc_m↑ | Sens↓ | Acc_n | Acc_m↑ | Sens↓ | Acc_n | Acc_m↑ | Sens↓ | Acc_n | Acc_m↑ | Sens↓ |
> | Kimi-K2.5 | **61.21** | **27.10** | 91.43 | 71.42 | 21.89 | 82.24 | 73.83 | 10.23 | 74.76 | 49.51 | 33.77 | 96.15 | 62.50 | 35.00 | 73.33 | 44.76 | 38.96 | 86.00 | 65.00 | 24.42 |
> | MiMo-VL-7B | **41.03** | **48.90** | 96.19 | 59.05 | 38.61 | 77.57 | 30.84 | 60.24 | 77.67 | 49.51 | 36.25 | 85.58 | 15.38 | 82.02 | 59.05 | 38.10 | 35.48 | 86.00 | 54.00 | 37.21 |
> | Gemini-3-Pro | **68.59** | **21.76** | 92.38 | 81.90 | 11.34 | 81.31 | 73.83 |9.20 | 88.35 | 63.11| 28.57 | 95.19| 67.31 | 29.29 | 79.05 | 50.48 | 36.14 | 90.00 | 75.00| 16.67 |
> | Human| **98.24** | **1.63** | 100.0 | 98.10 | 1.99 | 100.0 | 98.43 |1.57 | 99.03 | 97.09| 1.96 | 100.0| 98.08 | 1.92 | 100.0 | 100.0 | 0.00 | 100.0 | 98.00| 2.00 |
>
> As shown in table, even the strongest newly added model, Gemini-3-Pro, achieves an overall Acc_m of 68.59% with an MVI-Sensitivity of 21.76%, which still remains far below human performance (Acc_m = 98.24%, Sens = 1.63%).
>
> **Key Question: Language bias sanity check.** As described in Sec. 4.1, we randomly shuffle the answer options each time the LVLMs are queried to mitigate selection bias in MCQs. To further verify that language bias is limited, we evaluate a strong text-only LLM, Qwen3-30B-A3B, using only the question and answer options, without image input. As shown in the table below, the text-only accuracy (26.36%) is very close to random choice (25.00%), suggesting that language bias in MVI-Bench is minimal.
>
> **Table: Text-only language bias sanity check.** Accuracy (%) of Qwen3-30B-A3B with only question and options (no image input).
> | Setting|  Overall (%)|
> |--|--|
> | Qwen3-30B-A3B (Text-only) | 26.36 |
> | Random choice |  25.00 |

---

> > ### Author Rebuttal · Reviewer_EG62 · 2026-04-03
> >
> > Thank you for your updated results. I have no further concerns.

---

### Decision · Program_Chairs · 2026-04-30

**Decision:**

Accept (regular)

**Comment:**

This paper introduces MVI-Bench, the first benchmark to systematically evaluate the impact of misleading visual inputs on the robustness of LVLMs. This work fills an important gap in existing evaluation frameworks, featuring a scientifically designed benchmark, a clear taxonomy, and controllable data quality. Through comprehensive evaluations on 18 SOTA models, the paper reveals that current LVLMs exhibit significant vulnerability when facing visually misleading cues that humans can easily resolve, and provides insightful analysis and findings. Although there is room for improvement in dataset scale and the attribution analysis between perception and reasoning has certain limitations, the authors have effectively addressed the main concerns during the rebuttal through substantial supplementary experiments and appropriate refinements of their claims. This benchmark is expected to advance research on LVLM robustness and holds clear practical value and impact for the community.